# Multi-modal Learning via Slot-Guided Fine-grained Alignment with Pre-trained Uni-modal Models

## Abstract

Learning multi-modal representation with cross-modal correspondence often re-
lies on high quality multi-modal datasets with correspondence information.
Preparing multi-model datasets is costly, let alone together with the correspon-
dence information. Recently, many pretrained uni-modal models trained with
massive data have been made available, where each of them have their own set
of concepts captured via representation learning. Our idea to address the multi-
modal data scarcity challenge is to align a multi-modal model with uni-modal
models with fine-grained cross-modal correspondence. To this end, we propose a
multi-modal learning framework called slot-guided alignment (SGA) which uti-
lizes slot attention to decompose both the multi-modal and uni-modal represen-
tations into disentangled slots. The slots obtained from the pretrained uni-modal
models helps the associated concepts to be better aligned. As slot attention can be
applied to diverse model architectures, a wide range of pretrained models can be
leveraged. In addition, the disentangled slots from each modality allows similarity
to be measured among them, which in turn allows cross-modal correspondence
to be established at the slot level and enables the pretrained uni-modal models
to contribute to the multi-modal representation learning in a fine-grained man-
ner. To demonstrate the effectiveness of the SGA framework, we conduct experi-
ments using visual-text datasets for retrieval tasks and visual question answering,
and visual-audio datasets for classification tasks. We mainly enhance three base-
lines using Our SGA, and the results show a significant enhancement compared to
the vanilla baselines and the competitive results can be achieved even with much
smaller training dataset.

## 1 Introduction

Multi-modal learning aims to integrate information of diverse types, such as text, images, and audio,
to gain a comprehensive understanding of diverse modalities. Learning multi-modal representa-
tion with cross-modal correspondence properly established can allow more fine-grained integration,
which in turn should enhance the model accuracy. High quality multi-modal datasets with correspon-
dence information is often expected, which however is scarce as preparing large-scale multi-model
datasets is time-consuming and costly, let alone together with the correspondence information. Thus,
some multi-modal datasets provide only human annotation with coarse-grained correspondence in-
formation, such as pairing information of image and text. For more fine-grained correspondence
between regions in an image with words in a sentence, methods like OSCAR (Li et al., 2020b) and
VinVL (Zhang et al., 2021) leverage region-level representations obtained by object detectors, with
the predicted category labels providing the bridge to the corresponding words in the text. This ob-
ject detection-based methods enables models to learn more precise correspondences between image
and text, thereby improving downstream performance even with limited manually annotated data.
As integrating object detectors could be sub-optimal, LLaVa (Liu et al., 2023) further combines
a language-only LLM (text modality) to enrich detection results from the object detectors. These
works, while still specifically designed for certain modalities, demonstrate the potential of estab-
lishing fine-grained cross-modal correspondence automatically and leverage the "knowledge" for
enhancing multi-modal learning without additional annotation efforts.

Recently, many pretrained uni-modal models trained with massive data have been made available, where each of them have their own set of concepts captured via representation learning. This motivates the following questions: *Can we harness arbitrary pretrained uni-modal models and distill knowledge from them for multi-modal learning? Can fine-grained correspondence be established among multi-modal models and uni-modal models for enhancing multi-modal learning in a more fine-grained manner?*

We here propose slot-guided alignment (SGA) that leverages adaptive slot attention (Fan et al., 2024) to extract orthogonal vectors (slots) from representations, where each slot represents a disentangled semantic concept for the input representations. Our idea is to extend it to help fine-grained correspondence establishment among modalities, where the slots with similar concepts are learned to be aligned among the uni-modal and the multi-modal representations. The slots serve as the "interface" between the pretrained uni-modal models and the multi-modal model so that uni-modal models can transfer "knowledge" to multi-modal model during training. As shown in Figure 1, our slot-guided alignment (SGA) framework imposes adaptive slot attention for uni-modal and multi-modal representations, and then selects out the most similar slot pairs between uni-modal and multi-modal models. The slots are aligned by allowing the matched ones to learn and evolve via contrastive learning. As slot attention itself is generic and can be applied to the representations of diverse models, an arbitrary pretrained uni-modal model can be leveraged by our framework to provide such fine-grained correspondence, which means the "knowledge" comes from arbitrary uni-modal model can be leveraged to help multi-modal learning, and SGA generalizes correspondence establishment beyond text as in most of the related works.

To evaluate the effectiveness of our proposed SGA framework, we conducted extensive experiments across image-text, image-audio, and video-audio tasks. For the image-text task, we used ViLT (Kim et al., 2021) as the baseline model for encoding image and text, with ViT (Dosovitskiy et al., 2020) and BERT (Devlin et al., 2019) as pretrained uni-modal models. Results show that ViLT trained with SGA significantly outperforms the original ViLT across diverse datasets. We evaluated two baselines, OGM (Peng et al., 2022) and TSM-AV (Peng et al., 2022), on video-audio datasets. These baselines differ in their visual encoders: OGM processes single images, while TSM-AV handles frame sequences to capture temporal dynamics. For OGM, we utilized pretrained ResNet-18 (He et al., 2016), YOLO (Redmon et al., 2016), and PANN (Kong et al., 2020) to guide multi-modal learning. For TSM-AV, we leveraged pretrained TSM (Lin et al., 2019) and PANN. Improvements in audio-related tasks demonstrate that SGA can effectively generate fine-grained correspondences beyond text, potentially enhancing multi-modal learning across diverse modalities. Our ablation studies further reveal that SGA achieves state-of-the-art performance with fewer data samples, making it particularly valuable in data-scarce scenarios. Moreover, stronger pretrained uni-modal models help yield better performance, and the multi-modal model's performance scales with the number of pretrained uni-modal models used, highlighting SGA's ability to harness diverse uni-modal knowledge to mitigate data limitations.

## 2 RELATED WORKS

**Fine-grained Correspondence with Object Tags** The use of predicted object tags to establish correspondence between vision and language has been extensively explored in prior work. Zhou et al. (2020) explored this paradigm by concatenating object probability vectors with region features for visual-language pretraining, and Li et al. (2020b); Zhang et al. (2021) further extended this idea to simultaneously align object regions with both visual and word representations. LLava Liu et al. (2023) makes use of large language models to enrich the tags from the object detector for more detailed instructions. TUNA Qi et al. (2024) uses retrieval augmentation to generate more accurate instructions from large language models. Feinglass & Yang (2024) proposed semantic grounding to address misalignment of object proposals for vision-language tasks. Yi et al. (2024) supplemented simple tag information to get better performance. Most of the previous methods focus on text-related tasks. Our slot-guided alignment (SGA) framework extends the fine-grained correspondence generation from specific modalities (e.g., image-text) to general modalities via slots decomposed from arbitrary uni-modal models.

**Diverse Architectures for Multi-Modal Learning** Recent advances in multi-modal learning have explored diverse architectures. Late-fusion models like OGM Peng et al. (2022), CLIP Radford et al.

(2021) and MLA Zhang et al. (2024) employ deep uni-modal encoders with shallow cross-modal layers, aligning modalities in a shared embedding space. Early-fusion models such as ViLT Kim et al. (2021) use transformer encoders to mix modalities at the input level, enabling efficient cross-modal interaction. Other models, like OSCAR Li et al. (2020b), LLava Liu et al. (2023) and TUNA Qi et al. (2024), employ deep image encoders with deep cross-modal layers. Different from object detection-base methods (e.g. OSCAR and LLava), our slot-guided alignment (SGA) framework is suitable for the architectures mentioned above, which is a more flexible framework for fine-grained correspondence generation in multi-modal representation learning.

**Slot Attention for Multi-modal Learning** Slot attention Locatello et al. (2020) is a powerful mechanism for unsupervised object-centric representation learning, enabling dynamic decomposition of input features into semantically meaningful slots. Unicode Zheng et al. (2024) leverages discrete slots to represent visual and text tokens together to make them align. SlotFusion Han et al. (2025) uses slot attention to help image and audio understanding. CLS-3D Mushtaq et al. (2025) uses slot reweighting mechanism to fuse image and point cloud features. Kim et al. (2025) achieves performance sound source localization with slot attention to decompose image and audio into separating slots. In our work, we use slot attention for fine-grained correspondence generation, leveraging slot attention to decompose both uni-modal and multi-modal representations into disentangled slots, and then these slots help uni-modal and multi-modal establish fine-grained correspondence.

## 3 METHODS

Our proposed slot-guided alignment (SGA) framework, as shown in Figure 1, leverages adaptive slot attention (Fan et al., 2024) to identify disentangled slots (concepts) from representations generated by a multi-modal model or a set of pretrained uni-modal models and then reinforces the selected paired slots via a contrastive loss to align similar semantic slots between the uni-modal and multi-modal models in the same latent space. Though initial latent spaces of uni-modal and multi-modal models are different, the concepts from the uni-modal representations should exist in multi-modal representations as the input of the uni-modal model is part of the multi-modal model's. The SGA framework tries to identify these concepts using slots and align them between uni-modal and multi-modal models so that well pretrained uni-modal models could provide "knowledge" for multi-modal model during training. In the sequel, we first provide the problem formulation of SGA. Then we explain how SGA is learned to generate the slots. Finally, we present how SGA selects the appropriate slots and can enable the latent space alignment between the multi-modal model and the pretrained uni-modal models to achieve better multi-modal representation learning.

### 3.1 PROBLEM FORMULATION

Let $d = \{x_m\}_{m=1}^M$ denote a multi-modal sample with $M$ modalities (e.g., images, text, audio) where $x_m$ corresponds to the data of modality $m$. A multi-modal model $F$ takes $d$ as input and attains joint representations $H = F(d) \in \mathbb{R}^{N \times d}$. For each modality in $d$, we assume the availability of a pretrained uni-modal model $f_m$ which can generate uni-modal representations $h_m = f_m(x_m) \in \mathbb{R}^{N_m \times d}$. As these uni-modal models are pretrained independently on uni-modal data, their representations lie in their own latent spaces. Our goal is to bridge the representation discrepancy between multi-modal representations $H$ and these uni-modal representations $\{h_m\}_{m=1}^M$ so that learning of $H$ can be enhanced by leveraging the knowledge embedded in the uni-modal representations $\{h_m\}_{m=1}^M$.

The overall procedure is to first map $H$ into $K$ disentangled slots $\{\hat{s}_k\}_{k=1}^K$, and do the same for each modality in $\{h_m\}_{m=1}^M$ to obtain $\{s_k^m\}_{k=1}^{K_m}$ via adaptive slot attention. Because multi-modal model should output more concepts than single uni-modal model, we assume $K$ is bigger than $K_m$. For each pretrained uni-modal model $f_m$, SGA will select $K_m$ slot pairs with the highest similarity between the uni-modal models and multi-modal model, and then leverage contrastive loss to reinforce these pairs during training. So, the adaptive slot attention layer of the multi-modal model and the pretrained uni-modal models are learned in a coordinated manner which forces the generated slots to be in the same latent space.

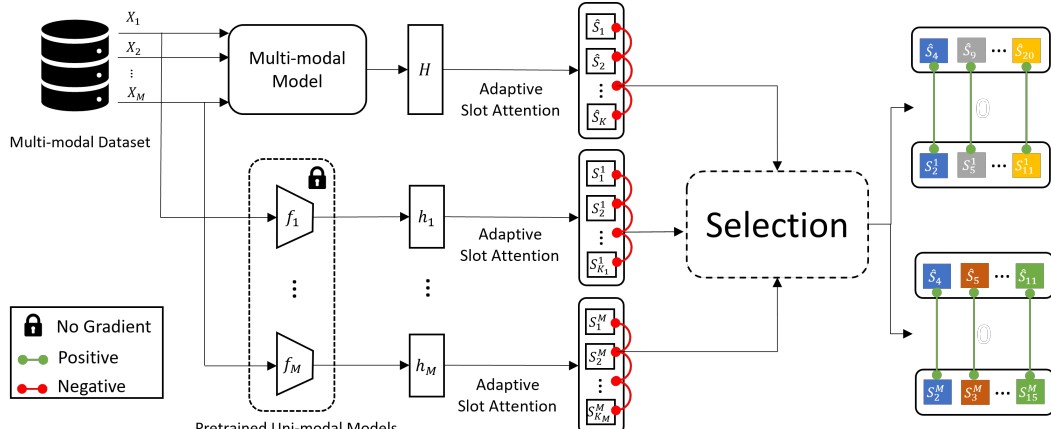

Figure 1: Slot-guided alignment (SGA) framework pipeline. The Multi-modal Model first maps the multi-modal sample into representation $H$, where the pretrained uni-modal models maps their corresponding modality samples into representations $h_m$. Then, slot attention is to decompose these representations into concepts $\{\hat{s}_k\}_{k=1}^{K}$ and $\{s_k^m\}_{k=1}^{K_m}$ and the contrastive loss will regard each intra-modal slot as a negative relation. Finally, it selects the most similar slots pairs based on cosine similarity between each uni-modal model and the multi-modal model, where the contrastive loss will regard these pairs as positive relations.

## 3.2 Adaptive Slots Generation

The SGA first define $L$ learnable slot embeddings for each uni-modal model and multi-modal model, represented as $S^m := \{s_k^m\}_{k=1}^{L} \in \mathbb{R}^{L \times d}$ and $\hat{S} := \{\hat{s}_k\}_{k=1}^{L} \in \mathbb{R}^{L \times d}$, where each embedding is sampled from a Gaussian distribution with learnable mean and variance parameters. These slot embeddings serve as initial representations that compete to capture distinct concepts from the input representation $H$ and $H_m$. Through an iterative attention-based refinement process, the slots act as queries that interact with the input representations (serving as keys and values) over $T$ iterations. This process results in $L$ refined slot embeddings for both the multi-modal model ($\hat{S}$) and each uni-modal model ($S^m$), effectively converting the representations into concepts (Appendix A for details).

Since samples contain varying numbers of concepts, a fixed number ($L$) of slots may result in void slots generation for slot attention. Inspired by Fan et al. (2024), SGA employs a lightweight selection network (binary classification), to identify non-void slots from $\hat{S}$ and $K_m$ non-void slots from $S^m$, yielding the final slot representations $\hat{S} := \{\hat{s}_k\}_{k=1}^{K} \in \mathbb{R}^{K \times d}$ and $S^m := \{s_k^m\}_{k=1}^{K_m} \in \mathbb{R}^{K_m \times d}$ (Appendix B for details).

## 3.3 Slot-Guided Alignment

Following the adaptive slot generation, the pairwise similarities between $\{\hat{s}_k\}_{k=1}^{K}$ and $\{s_k^m\}_{k=1}^{K_m}$ are calculated to provide the correspondence score for the alignment among the slots of the multi-modal model and the uni-modal models. To force the slots from the uni-modal models and the multi-modal model in the same space, SGA computes the dot product similarity of slot pairs between them and encourages the pairs with the highest similarities via a contrastive loss. The slots with similar concepts are thus aligned iteratively during training.

**Inter-model Slot Similarity**  Given the slot embeddings $\{\hat{s}_k\}_{k=1}^{K}$ for the multi-modal representation and $\{s_k^m\}_{k=1}^{K_m}$ for each uni-modal representation, we compute the pairwise cosine similarity scores for each modality $m$:

$$\mathbf{C}^m[k,l] = \frac{\langle \mathbf{s}_k^m, \hat{\mathbf{s}}_l \rangle}{|\mathbf{s}_k^m||\hat{\mathbf{s}}_l|} \tag{1}$$

where $C^m \in \mathbb{R}^{K_m \times K}$ is the slot similarity matrix for modality $m$. This matrix captures the strength of correspondence between $\{\hat{s}_k\}_{k=1}^{K}$ and $\{s_k^m\}_{k=1}^{K_m}$, with a higher value indicating stronger semantic similarity.

**Competitive Slot Selection**   Due to potentially incomplete information in the uni-modal sample $x_m$ compared to the multi-modal sample $d$, there might only be part of the concepts in $d$ which can be identified. The SGA only reinforces $K_m$ most similar pairs from the slot similarity matrix $C^m$, where this operation respects our assumption that multi-modal representations contain more concepts than uni-modal representations ($K > K_m$).

$$\mathcal{N}^m = \text{top-}K_m \ \mathbf{C}^m \tag{2}$$

where top-$K_m$ means selecting the $K_m$ highest scores from the matrix. This competitive selection ensures that only the most salient inter-model slot pairs will be reinforced by the subsequent contrastive learning.

**Contrastive Loss**   The SGA adopts a contrastive loss which pulls aligned slots within $\mathcal{N}^m$ closer and at the same time pushes the slots of intra-modal slots apart to encourage extraction of distinct concepts. The loss function is defined as:

$$\mathcal{L}_{\text{cont}} = -\frac{1}{M} \sum_{m=1}^{M} \log \frac{\sum_{c \in \mathcal{N}^m} \exp(c)}{\sum_{k \neq l} \exp(\frac{\langle \mathbf{s}_k^m, \mathbf{s}_l^m \rangle}{|\mathbf{s}_k^m||\mathbf{s}_l^m|}) + \sum_{k \neq l} \exp(\frac{\langle \hat{\mathbf{s}}_k, \hat{\mathbf{s}}_l \rangle}{|\hat{\mathbf{s}}_k||\hat{\mathbf{s}}_l|})}. \tag{3}$$

The numerator is to encourage higher similarity between the aligned slot pairs so that different models will be projected into the same latent space, whereas the denominator is to discourage similarity among the intra-modal slots. The denominator guides the adaptive slot attention layer to generate more slots for the multi-modal model than for uni-modal models. The selection of $K_m$ inter-modal pairs leads to multiple intra-modal slots reinforced to the same multi-modal slot while $K_m$ is larger than $K$, which compromises the contrastive loss. With the constraint of the contrastive loss, the adaptive slot attention will output more slots for the multi-modal model than uni-modal models.

**Retrieval Loss**   The SGA utilizes retrieval loss to align joint representations $H$, comprising $M$ sets of embeddings denoted as $H_i^{(m)}$ for the $m$-th modality of the $i$-th sample, within a shared embedding space. The objective is to maximize the similarity between embeddings of the same sample across modalities while minimizing the similarity across different samples. The similarity $\text{sim}(H_i^{(m)}, H_j^{(z)})$ between embeddings of modalities $m$ and $z$ for samples $i$ and $j$ is computed using transport distance (Kolouri et al., 2017). The retrieval loss is defined as:

$$\mathcal{L}_{\text{retri}} = -\frac{1}{B} \sum_{b=1}^{B} \log \frac{\sum_{m \neq 1} \exp(sim(H_b^{(1)}, H_b^{(m)}))}{\sum_{i \neq b} \sum_{m \neq 1} \exp(sim(H_b^{(1)}, H_i^{(m)}))} \tag{4}$$

where $B$ is the batch size. This loss enables region-level alignment by working in tandem with slot attention, which organizes representations from different modalities into shared slots corresponding to semantic regions. The inter-modal retrieval loss ensures that these grouped representations within the same slot are semantically consistent across modalities, as only such alignments minimizes the loss value.

## 4   REGION LEVEL ALIGNMENT VISUALIZATION

Our slot-guided alignment (SGA) framework reinforces the aligned slots between the multiple uni-modal models and the multi-modal model. To gain insight whether the aligned slots are in fact representing semantically consistent concepts across different modalities, we can choose and highlight

The Sonic Headquarters building on the right, next to the river in Bricktown.

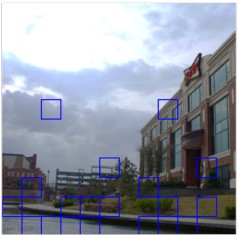 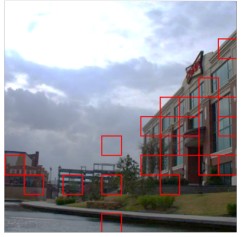

Little Indian girl touching a bronze statue of a growling tiger or lion in the grounds.

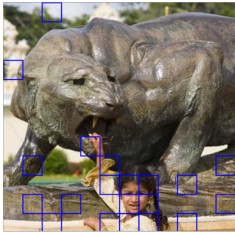 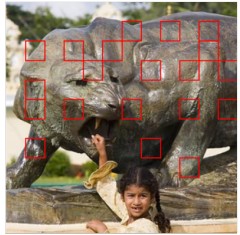

Lettuce in wooden wine box - and a classy coffee table from IKEA repurposed as a plant stand

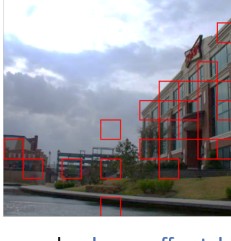 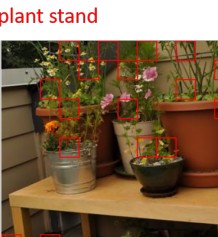

Mountain in the other direction, back of the house

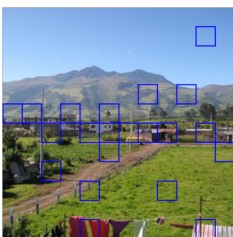 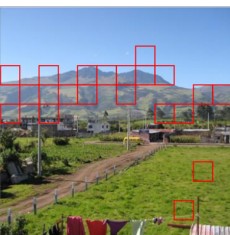

Figure 2: Slots visualization. We visualize the slots from uni-modalities aligning with the same multi-modal slots. Each sub-figure represents an image-text sample from the SBU dataset. The red (blue) patches indicate the top-20 contributions to the image slot that aligns with a multi-modal slot. The red (blue) words in the text denote the top-3 contributions to the text slot that aligns with the same multi-modal slot.

the slots from different uni-modal representations which align with the same slot from multi-modal representation. Specifically, the tokens (e.g., words in text and patches in image) with the highest attention weight $A_{i,j}$ contributing to the uni-modal slots can be highlighted. Figure 2 depicts four different cases. For each case, we visualize the image and text modalities by highlighting the top 20 patches and the top 3 words contributing to the uni-modal slots aligning with the same slot from multi-modal representation. From the figures, we can see that the semantics of the patches and words in the same color are in fact consistent to each other. For instance, for the upper left case, one extracted concept is obviously "Headquarter building on the right" (red) with both words and bounding boxes showing coherent indication. Another one is "next to the river" (blue). And similarly, we can notice the coherent indication by the corresponding words and bounding boxes (blue).

## 5 EXPERIMENTS

### 5.1 OVERVIEW

To show effectiveness of SGA, we conduct a comprehensive performance evaluation over image-text, image-audio and video-audio task. For image-text task, we use ViLT (Kim et al., 2021) as multi-modal model to encode image and text and ViT (Dosovitskiy et al., 2020) and BERT (Devlin et al., 2019) as pretrained uni-modal models.The ViLT was first pretrained using four standard datasets: MSCOCO Lin et al. (2014), Visual Genome Krishna et al. (2017), SBU Captions Ordonez et al. (2011), and Google Conceptual Captions Sharma et al. (2018) in SGA framework. And the pretrained models are further finetuned in MSCOCO (Lin et al., 2014), Flickr30k (Plummer et al., 2015), NLVR (Suhr et al.) and VQA2 (Goyal et al., 2017) for retrieval and question answering tasks.

We evaluate image–audio and video–audio tasks using two baselines (OGM (Peng et al., 2022) and TSM-AV (Peng et al., 2022)) on the same three datasets: CREMA-D (Cao et al., 2014), Kinetics-Sounds (KS) (Arandjelovic & Zisserman, 2017), and VGGSound (Chen et al., 2020a). The key difference lies in their visual encoders: OGM processes a single image per sample, whereas TSM-AV ingests sequences of frames to capture temporal relationships. For OGM, we leverage

pretrained ResNet-18 (He et al., 2016), YOLO (Redmon et al., 2016), and CNN14 (Kong et al., 2020) to guide multi-modal learning; for TSM-AV, we use pretrained TSM (Lin et al., 2019) and CNN14. Implementation details can be found in Appendix.

## 5.2 VISUAL-TEXT TASKS

### 5.2.1 TEXT-IMAGE RETRIEVAL

The retrieval task is to evaluate the multi-modal model in distinguishing paired data from a cluster of unpaired data. For text (image) retrieval, the multi-modal model is to select the most paired texts (images) given the input image (text). We tested the model in two situations: i) Zero-shot retrieval refers to the setting where the model after pre-training is not fine-tuned; ii) Fine-tuned retrieval refers to the setting with fine-tuning using the corresponding datasets after the pre-training stage. The evaluation metrics include: R@$n$ the percentage of queries where the top-$n$ retrieved result (either image or text) is correct. The results of performance comparison on zero-shot retrieval and fine-tuned retrieval are summarized in Tables 1 and 2. We group the methods using object detector for image encoding (labeled as "region" in the first column), and those not using object detector (labeled as "linear") separately. The last two rows of the tables report the performance of the models with same architecture and training steps.

As shown in the tables, ViLBERT (Lu et al., 2019), Unicoder (Li et al., 2020a), UNITER (Chen et al., 2020b), imgBERT (Qi et al., 2020), OSCAR (Li et al., 2020b) and VinVL (Zhang et al., 2021) leverage object detector as their viusal encoder, so they need much more inference time. ViLT get rid of object detector to gain fast inference time, but it suffer from performance loss without the help of well pretrained uni-modal model. ViLT trained by our SGA framework achieves competitive retrieval results on Flickr30K and MSCOCO and outperforms the vanilla ViLT despite pre-training on a smaller dataset without increasing the inference time. For zero-shot retrieval, our model surpasses ViLT from 3.6 to $5.5\%$. For fine-tuned retrieval, the improvement is from $1.5$ to $4.1\%$.

Table 1: Pretrained SGA version ViLT on Flickr30k and MSCOCO for zero-shot text-image retrieval. The third column shows the number of pre-training samples in millions (m). The fourth column displays inference time in milliseconds (ms).

| Visual Encoder | Model | Samples (m) | Time (ms) | Zero-Shot Text Retrieval | | | | Zero-Shot Image Retrieval | | | |
| | | | | Flickr30k (1K) | | MSCOCO (5K) | | Flickr30k (1K) | | MSCOCO (5K) | |
| | | | | R@1 | R@5 | R@1 | R@5 | R@1 | R@5 | R@1 | R@5 |
| Region | ViLBERT | 3 | 900 | 58.2 | 84.9 | - | - | 31.9 | 61.1 | - | - |
| | Unicoder | 4 | 925 | 64.3 | 85.8 | 54.4 | 82.8 | 48.4 | 76.0 | 43.4 | 76.0 |
| | UNITER | 10 | 900 | 80.7 | 95.7 | - | - | 66.2 | 88.4 | - | - |
| | ImgBERT | 14 | 925 | 70.7 | 90.2 | 44.0 | 71.2 | 54.3 | 79.6 | 32.3 | 59.0 |
| Linear | ViLT-B | 10 | 15 | 69.7 | 91.0 | 53.4 | 80.7 | 51.3 | 79.9 | 37.3 | 67.4 |
| | **ViLT-B (SGA)** | **8** | **15** | **75.2** | **94.6** | **57.5** | **84.6** | **55.8** | **83.5** | **42.3** | **72.0** |

Table 2: Finetune SGA version ViLT on Flickr30k and MSCOCO for text-image retrieval. The third column displays inference time in milliseconds (ms).

| Visual Encode | Model | Time (ms) | Text Retrieval | | | | Image Retrieval | | | |
| | | | Flickr30k (1K) | | MSCOCO (5K) | | Flickr30k (1K) | | MSCOCO (5K) | |
| | | | R@1 | R@5 | R@1 | R@5 | R@1 | R@5 | R@1 | R@5 |
| Region | ViLBERT | 920 | - | - | - | - | 58.2 | 84.9 | - | - |
| | Unicoder | 925 | 86.2 | 96.3 | 62.3 | 87.1 | 71.5 | 91.2 | 48.4 | 76.7 |
| | UNITER | 900 | 85.9 | 97.1 | 64.4 | 87.4 | 72.5 | 92.4 | 50.3 | 78.5 |
| | OSCAR | 900 | - | - | 70.0 | 91.1 | - | - | 54.0 | 80.8 |
| | VinVL | 650 | - | - | 74.6 | 92.6 | - | - | 58.1 | 83.2 |
| Linear | ViLT-B | 15 | 81.4 | 95.6 | 61.8 | 86.2 | 61.9 | 86.8 | 41.3 | 72.0 |
| | **ViLT-B (SGA)** | **15** | **85.3** | **97.1** | **63.4** | **87.7** | **65.9** | **89.9** | **45.4** | **75.8** |

### 5.2.2 Visual Question Answering

Table 3: Visual question answering. The third column displays inference time in milliseconds (ms).

| Visual Encoder | Model | Time (ms) | VQAv2 | NLVR2 |
|---|---|---|---|---|
| Region | ViLBERT | 920 | 70.55 | - |
| | UNITER | 900 | 72.70 | 75.85 |
| | OSVAR | 900 | 73.16 | 78.07 |
| Linear | ViLT | 15 | 70.33 | 74.41 |
| | **ViLT (SGA)** | 15 | **73.21** | **77.8** |

Table 4: SGA version OGM with different number of slots $K$. KS denotes Kinetics-Sounds.

| $K$ | CREMA-D | KS | VGGSound |
|---|---|---|---|
| 6 | 61.4 | 63.1 | 50.8 |
| 8 | 61.8 | 63.5 | 51.3 |
| 10 | 62.3 | 64.4 | 51.6 |
| 12 | 62.5 | 64.3 | 51.4 |
| 14 | 62.3 | 64.2 | 51.5 |

In visual question answering (VQA) tasks, it requires the model to capture finer-grained correspondence to get good performance as the model needs to figure out the related region mentioned in the text from the image. We further finetune the pretrained ViLT for the retrieval task and evaluate it using two VQA datasets VQAv2 and NLVR2 as shown in Table 3. The VQAv2 and NLVR2 tasks ask for answers given pairs of an image and a question in natural language. And it is a common practice to convert the task to a classification task. Similar to retrieval task, as the region methods embed the object detector inside their models, they usually requires more time ($\sim$900) than early fusion models (linear) like ViLT. Yet, the knowledge distilled from the pretrained uni-modal models make them achieve higher performance. Our SGA enables early fusion models to leverage pretrained uni-modal models so that its performance is comparable with that of the object detector embedded models.

### 5.3 Visual audio tasks

For the visual-audio classification task, the multi-modal model is expected to predict the class of the video and audio input. As seen in Table 5, to show that our SGA works both for video and image modalities, we leverage our SGA in two models OGM and TSM-AV (Peng et al., 2022) for video-audio classification task. our proposed SGA framework gains performance enhancement in term of accuracy by 2.3–7.9% on CREMA-D, Kinetics-Sounds and VGGSound. Interestingly, TSM-AV outperforms OGM with the help of our SGA even though TSM-AV has lower performance in its vanilla version. We anticipate that TSM-AV utilizes TSM that could capture both spatial and temporal relationships in the video. So the under-explored expressiveness is higher than OGM using ResNet-18 that could only capture spatial relationships. These improvements stem from our framework that enables the model to leverage "knowledge" from other well pretrained uni-model models.

Table 5: OGM and TSM-AV experiments in SGA.

| Method | CREMA-D | KS | VGGSound |
|---|---|---|---|
| TSM-AV | 55.4 | 60.3 | 48.8 |
| OGM | 59.0 | 62.2 | 49.6 |
| **OGM (SGA)** | 62.5 | 64.3 | 51.9 |
| **TSM-AV (SGA)** | **63.3** | **65.4** | **52.8** |

Table 6: OGM in SGA with different uni-modal learning rate (ULR).

| ULR | CREMA-D | KS | VGGSound |
|---|---|---|---|
| 1e-5 | 57.7 | 59.9 | 50.9 |
| 1e-6 | 60.5 | 62.3 | 51.3 |
| 1e-7 | 61.7 | 63.2 | 51.4 |
| 0 | 62.5 | 64.5 | 51.9 |

### 5.4 Ablation Study

To test if the performance of our framework is sensitive towards the size of the training set, we conducted a comparison study and trained ViLT using dataset of different sizes. As shown in Table 7, even if the training samples are reduced to only 20%, the multi-modal models trained using our framework can still maintain well their performance and achieve comparable results in zero-shot text and image retrieval tasks on Flickr30k and MSCOCO datasets.

We conduct a sensitivity analysis on the hyper-parameter $K$ in slot attention, based on OGM settings. The parameter $K$ represents the maximum number of concepts (slots) that adaptive attention can output. Since different datasets have varying maximum $K$ values, $K$ must be at least as large as the

Table 7: Samples sensitivity on Flickr30k and MSCOCO for Text-Image retrieval. The first row denotes vanilla ViLT-B pretrained with 10 millions (m) samples. The rest of the rows denote ViLT-B pretrained with different sizes of samples using SGA.

| Methods | Samples (m) | Zero-Shot Text Retrieval | | | | | | Zero-Shot Image Retrieval | | | | | |
|---|---|---|---|---|---|---|---|---|---|---|---|---|---|
| | | Flickr30k (1K) | | | MSCOCO (5K) | | | Flickr30k (1K) | | | MSCOCO (5K) | | |
| | | R@1 | R@5 | R@10 | R@1 | R@5 | R@10 | R@1 | R@5 | R@10 | R@1 | R@5 | R@10 |
| ViLT-B | 10 | 69.7 | 91.0 | 96.0 | 53.4 | 80.7 | 88.8 | 51.3 | 79.9 | 87.9 | 37.3 | 67.4 | 79.0 |
| ViLT-B (SGA) | 2 | 69.2 | 91.2 | 95.7 | 53.5 | 80.4 | 88.3 | 52.0 | 79.5 | 88.1 | 36.8 | 65.3 | 80.1 |
| | 4 | 72.7 | 93.0 | 96.8 | 55.8 | 83.5 | 89.8 | 54.3 | 81.9 | 91.7 | 40.3 | 70.4 | 81.8 |
| | 6 | 73.1 | 93.8 | 97.0 | 56.9 | 83.7 | 89.9 | 54.8 | 82.9 | 91.9 | 41.2 | 71.4 | 82.3 |
| | 8 | 75.2 | 94.6 | 97.1 | 57.5 | 84.6 | 90.6 | 55.8 | 83.5 | 92.8 | 42.3 | 72.0 | 83.1 |

maximum number of concepts in any sample from the dataset. As shown in Table 4, the optimal $K$ is approximately 10, and increasing it beyond this value has minimal impact on performance in the CREMA-D, Kinetics-Sounds (KS), and Kinetics-Sounds (KS) datasets. This is due to adaptive slot generation, which dynamically adjusts the number of slots to fit each sample as long as the maximum slot setting exceeds the dataset's requirements.

We investigated the uni-modal learning rate to assess the expressiveness of slot attention in generating slots within a consistent representational space. Experiments were conducted by adjusting the learning rate of the pretrained uni-modal model. As presented in Table 6, the findings highlight the expressiveness of slot attention, demonstrating that it can effectively generate slots in the same space, with optimal performance achieved only when the uni-modal learning rate is fixed.

We utilized PANN (CNN10 and CNN14), ResNet (R18 and R50), and YOLOv5 (Y) as pretrained uni-modal models to distill knowledge, enhancing the Object-guided Multimodal (OGM) framework for video-audio classification tasks using SGA. According to Table 8, CNN14 and R50 achieved the best performance, indicating that stronger uni-modal models enhance multi-modal learning in SGA. Furthermore, Table 9 reveals that the combination of CNN10, R18, and Y yields the highest performance, demonstrating that SGA's multi-modal learning scales with the number of uni-modal models employed. Also, audio model (first and third row) makes more contribution to VGGSound, possibly because audio dominates the classification task in VGGSound. This is expected, as leveraging stronger and more diverse uni-modal models allows SGA to utilize a broader range of datasets, thereby improving multi-modal learning outcomes.

Table 8: OGM with pretrained uni-modal models of different capability.

| Audio | Vision | CREMA-D | KS | VGG |
|---|---|---|---|---|
| CNN10 | R18 | 61.0 | 62.8 | 51.2 |
| CNN10 | R50 | 61.0 | 62.8 | 51.2 |
| CNN14 | R18 | 61.5 | 63.3 | 51.5 |
| CNN14 | R50 | 62.5 | 64.3 | 51.9 |

Table 9: OGM with different combinations of pretrained uni-modal models.

| Audio | Vision | CREMA-D | KS | VGG |
|---|---|---|---|---|
| - | - | 51.7 | 59.8 | 48.9 |
| - | R | 60.5 | 62.4 | 49.0 |
| CNN10 | - | 61.0 | 62.8 | 51.2 |
| CNN10 | R18 | 61.5 | 63.3 | 51.5 |
| CNN10 | R18 & Y | 62.5 | 64.3 | 51.9 |

## 6 CONCLUSION

We propose the slot-guided alignment framework for fine-grained correspondence generation, that enables knowledge to be effectively distilled from arbitrary pretrained uni-modal models for better multi-model learning. Extensive experiments were conducted across visual-language and visual-audio tasks, and the results demonstrated that the proposed framework can effectively enhance vanilla models in these tasks, hinting its potential to be applied to diverse modalities. For limitations, the effectiveness of the framework for more modalities (e.g., point cloud) is to be evaluated. Also, the theoretical properties for the slot-guided alignment mechanism should be more carefully studied.

ETHICS STATEMENT

All authors of this submission have read and agree to adhere to the ICLR Code of Ethics, available at `https://iclr.cc/public/CodeOfEthics`. We confirm that our work complies with the ethical guidelines outlined therein. Specifically, this study does not involve human subjects, sensitive data, or potentially harmful applications.

REPRODUCIBILITY STATEMENT

To ensure the reproducibility of our results, we have provided comprehensive details across the main text and appendix. The novel algorithm proposed in Section 3 is fully described. The main source code used for experiments will be available after reviewing. The datasets used are publicly accessible. All experiments were conducted using standard, open-source libraries. These resources collectively enable full reproducibility of our findings.

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

# A   SLOT ATTENTION

*Slot Initialization*: We first define $L$ learnable slot embeddings $S^m := \{s_k^m\}_{k=1}^L \in \mathbb{R}^{L \times d}$ for each uni-modal model and $\hat{S} := \{\hat{s}_k\}_{k=1}^L \in \mathbb{R}^{L \times d}$ for the multi-modal representation. Each slot embedding is initialized by sampling from a Gaussian distribution with learnable mean and variance. These slots will then compete and refine iteratively to explain different concepts of the input representation.

*Iterative Attention Refinement*: Given the input representation $H \in \mathbb{R}^{N \times d}$, slot attention refines the slots through $T$ iterations of attention operations with the slots $S$ as queries and the input representation $H$ as the keys and values:

$$\mathbf{Q} = \mathbf{W}_q \mathbf{S}, \quad \mathbf{K} = \mathbf{W}_k \mathbf{H}, \quad \mathbf{V} = \mathbf{W}_v \mathbf{H} \tag{5}$$

where $\mathbf{W}_q, \mathbf{W}_k, \mathbf{W}_v \in \mathbb{R}^{d \times d}$ are the learned projection matrices. The attention weights of the $i$-th input feature vector in $H$ are normalized using a softmax with respect to $K$ slots:

$$attn_{i,j} = \frac{e^{M_{i,j}}}{\sum_{l=1}^K e^{M_{i,l}}}, \quad where \quad M := \frac{1}{\sqrt{d}} K Q^T \in \mathbb{R}^{N \times K}. \tag{6}$$

The slot embedding is then computed as a weighted mean over the $N$ input feature vectors:

$$\mathbf{S} = A^T V \in \mathbb{R}^{L \times D}, \quad where \quad A_{i,j} := \frac{attn_{i,j}}{\sum_{l=1}^N attn_{l,j}}. \tag{7}$$

This process yields $L$ slot embeddings $\{\hat{s}_k\}_{k=1}^L$ for the multi-modal model, and also $L$ slot embeddings $\{s_k^m\}_{k=1}^L$ for each uni-modal model.

# B   ADAPTIVE SLOT ATTENTION

To be specific, we denote $S \in \mathbb{R}^{K \times D}$. A light weight neural network $h_\theta : \mathbb{R}^D \to \mathbb{R}^2$ is used to predict the keep/drop probability of each slot individually:

$$\pi = \text{Softmax}(h_\theta(S)) \in \mathbb{R}^{K \times 2}, \tag{8}$$

where $\pi_{i,0}$ denote the soft probability to drop the $i$-th slot, while $\pi_{i,1}$ denote the soft probability to keep the $i$-th slot. By applying the Gumbel-Softmax with Straight-Through Estimation **?** on the probability dimension and take the last column, we get the hard decision slot mask $Z$:

$$Z = \text{GumbelSoftmax}(\pi)_{.,1}. \tag{9}$$

Here the column $(\cdot)$ denotes all rows, and 1 denotes the specific column we extract. Since Gumbel-Softmax generate onehot vector, take the column we get $K$-dimensional zero-one mask $Z = (Z_1, \cdots, Z_K) \in \{0, 1\}^K$.

# C   IMPLEMENTATION DETAILS

## C.1   DATASET STATISTIC

We can only collect part of the datasets GCC and SBU due to URL expiration or access limitation.

## C.2   EARLY-FUSION MULTI-MODAL ARCHITECTURE: VILT ALIGNED WITH VIT & BERT

We utilize ViLT  Dosovitskiy et al. (2020) which splits images into 32-pixel patches and converts both patches and text into tokens, and following ViLT-B default setting. We use pretrained uni-modal models ViT Dosovitskiy et al. (2020) and BERT  Devlin et al. (2019) to guide multi-modal representation learning via slot alignment. The ViT was trained on ImageNet  den (2009) for a classification task, and the BERT was trained on BookCorpus Zhu et al. (2015) and English Wikipedia

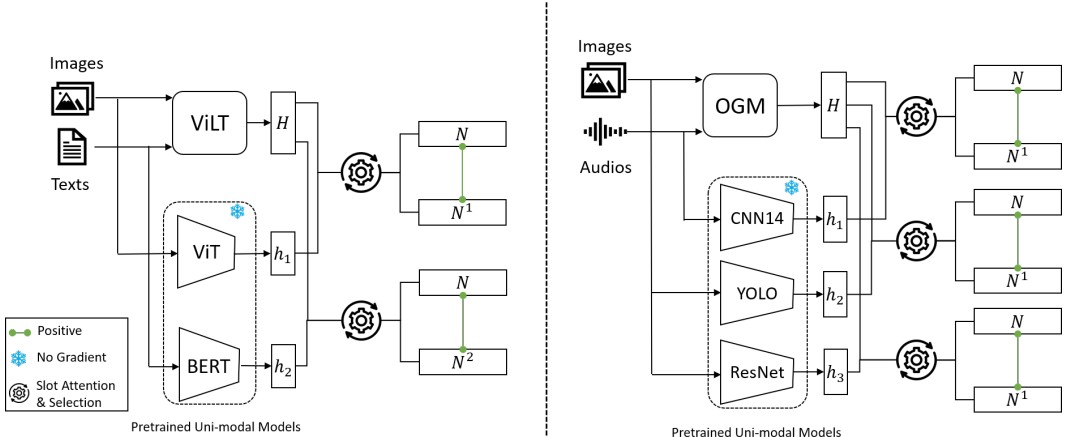

Figure 3: Two multi-modal architectures. On the left, early-fusion model ViLT processes image and text into multi-modal representations $H$, where pretrained ViT and BERT generate uni-modal representations $\{h_m\}_{m=1}^2$. On the right, late-fusion model OGM maps image and audio into multi-modal representations $H$, where pretrained CNN14, YOLO and ResNet generate uni-modal representations $\{h_m\}_{m=1}^3$. The slot attention and selection as illustrated in Figure 1 are then used to align the similar slots between uni-modalities and multi-modality.

Table 10: Dataset statistics.

| Dataset | # I | # T | # V | # pairs |
|---|---|---|---|---|
| MSCOCO | 113K | 567K | - | 567K |
| VG | 108K | 5.41M | - | 5.41M |
| GCC | 3.01M | 3.01M | - | 1.7M |
| SBU | 867K | 867K | - | 200K |
| Flickr30K | 31K | 159K | - | 159K |
| CREMA-D | - | - | 7K | 7K |
| Kinetics-sounds | - | - | 19K | 19K |
| VGGSound | - | - | 200K | 19K |

Xu & Lapata (2019). We utilized the AdamW optimizer with a base learning rate set at $1e^{-4}$ and a weight decay of $1e^{-2}$. We incorporate a linear warmup phase for the initial $10\%$ of training steps to ensure gradual adaptation. The model went through $25,000$ steps, 20 slots and selection $r = 5$ for pre-training with a batch size of 256, followed by $5,000$ steps for fine-tuning on the same hardware configuration for the downstream tasks.

### C.3 LATE-FUSION MULTI-MODAL ARCHITECTURE: OGM ALIGNED WITH RESNET, YOLO AND CNN14

Following OGM Peng et al. (2022), we utilized two ResNet-18 He et al. (2016) to encode video and audio and then a fully connected layer to merge the encoded video and audio representations into a multi-modal representation. An image is sampled from the video as input for the visual ResNet-18 and the audio input is transformed into frequency representation by Fast Fourier Transform (FFT) as input for the audio ResNet-18. For the uni-modal models, we adopted pretrained ResNet-18 He et al. (2016), YOLO Redmon et al. (2016) and CNN14 Kong et al. (2020). ResNet-18 was trained on ImageNet den (2009) for a classification task, YOLO was trained on MSCOCO Lin et al. (2014) for object detection task, and CNN14 was trained on AudioSet Gemmeke et al.. We utiliz the AdamW optimizer with a base learning rate set at $1e^{-4}$ and a weight decay of $1e^{-2}$. We incorporate a linear warmup phase for the initial $10\%$ of training steps to ensure gradual adaptation. The late-fusion model was also trained for $5,000$ steps with a batch size of $128$, specifically targeting the downstream tasks.

