# OpenReview forum: "Multi-modal Learning via Slot-Guided Fine-grained Alignment with Pre-trained Uni-modal Models"
_ICLR.cc/2026/Conference — ICLR 2026 Conference Withdrawn Submission_

### Official Review · Reviewer_4tF2 · 2025-10-24

**Soundness:** 2
**Presentation:** 2
**Contribution:** 2
**Rating:** 2
**Confidence:** 4

**Summary:**

This paper proposes a new framework to address the limitation of available multimodal data by transferring knowledge from unimodal pretrained models. The approach employs a slot-guided alignment mechanism, which leverages slot attention to decompose both multimodal and unimodal representations into disentangled slots. These slots are then aligned to a shared latent space to facilitate effective knowledge transfer to the multimodal model.

**Strengths:**

- The idea of transferring knowledge from pretrained unimodal models to multimodal models is interesting. If effective, it can substantially reduce the data requirements for training robust multimodal models with good generalizability.

- The use of slot attention to align different feature spaces into a common latent space is innovative. By decomposing representations into disentangled slots, semantically similar slots across modalities may be successfully aligned through contrastive learning.

- The paper is generally well written and easy to follow, with clear structure and motivation.

**Weaknesses:**

- In theory, any pretrained unimodal models can be utilized. However, when the two unimodal models differ significantly in their output feature spaces, it is unclear whether the slot attention mechanism can still effectively reduce these discrepancies. Additional experiments are needed to evaluate the impact of the chosen unimodal models.

- There now exist large-scale multimodal benchmark datasets and models, such as ImageBind [1] and VAST [2], that support effective multimodal representation learning. The authors should include comparisons and discussions related to these state-of-the-art methods. The current experiments primarily focus on improving ViLT using the proposed approach, leaving it unclear how much additional performance gain can be achieved when integrated with more advanced multimodal frameworks such as ImageBind or VAST.

- The proposed method seems most beneficial in scenarios where multimodal training data is limited. The authors may consider discussing potential applications in domains where large-scale multimodal data collection is challenging, to better motivate the usefulness of the approach.

[1] ImageBind: One Embedding Space To Bind Them All. CVPR, 2023.

[2] VAST: Vision-Audio-Subtitle-Text Omni-Modality Foundation Model and Dataset. NeurIPS 2023.

**Questions:**

- Can the proposed approach further improve more recent and advanced multimodal models?
- Can the proposed method be extended for more than two modalities?

---

### Official Review · Reviewer_f8Zj · 2025-10-27

**Soundness:** 3
**Presentation:** 3
**Contribution:** 2
**Rating:** 4
**Confidence:** 5

**Summary:**

This paper introduces a framework called Slot-Guided Alignment (SGA) for multi-modal learning.
In this paper, authors solves the problem of limited fine-grained cross-modal correspondence and the scarcity of large-scale annotated multi-modal datasets.
SGA uses pre-trained uni-modal models and uses Slot Attention to decompose both multi-modal and uni-modal representations into disentangled slots, which serve as semantic concepts.
These slots are aligned through competitive selection and contrastive learning, enabling knowledge transfer from uni-modal models to the multi-modal model.
An adaptive slot selection mechanism ensures that only meaningful slots are retained, and a retrieval loss based on optimal transport further enforces consistency across modalities.
Experiments on image-text retrieval, visual question answering, and visual-audio classification show that SGA performs better over baseline models while maintaining efficiency.
Ablation studies confirm its robustness to reduced training data and show that stronger and more diverse uni-modal models enhance results, proving its potential for scaling and effectiveness.

**Strengths:**

## Originality
This paper introduces Slot-Guided Alignment (SGA) as a mechanism to leverage pre-trained uni-modal models for multi-modal learning through slot-level alignment.
While Slot Attention itself is not new, its use as an interface for fine-grained cross-modal correspondence combined with adaptive slot selection and contrastive learning is a novel approach to the problem.
This approach overcomes a limitation in prior models by removing its reliance on object detectors, and applies to audio and video modalities, not just text.
This combination of ideas addresses a gap in existing methods that typically focus on coarse-grained alignment or modality-specific solutions.

## Quality
This technical design is robust and clearly motivated.
This paper provides a clear problem formulation, describes the adaptive slot generation process, and integrates competitive slot selection with contrastive loss to ensure semantic consistency.
Experiments are conducted across multiple tasks—image-text retrieval, visual question answering, and visual-audio classification—using established baselines.
Ablation studies on slot number, uni-modal learning rate, and model strength support the claims about scalability and robustness. However, the evaluation is limited by the lack of comparison with the latest state-of-the-art models.

## Clarity
This paper is presented clearly overall.
The motivation for addressing data scarcity and fine-grained alignment is clearly explained, and the pipeline is illustrated with helpful figures.
Mathematical formulations for slot similarity, contrastive loss, and retrieval loss are provided in detail, and implementation specifics are included in the appendix.
Some sections, such as the theoretical properties of slot-guided alignment, are noted as future work, and the discussion of limitations is brief, but overall the narrative is coherent and accessible.

## Significance
The proposed framework has practical value because it allows multi-modal models to use existing uni-modal resources without new annotations, which is valuable in data-scarce scenarios.
The ability to achieve competitive performance with reduced training data and maintain efficiency by avoiding object detectors indicates its potential for real-world deployment.
While the improvements over baselines are consistent, the lack of direct comparison with modern vision-language architectures reduces the perceived impact.

**Weaknesses:**

## Originality
This paper claims novelty by introducing SGA, but the core components—Slot Attention, contrastive learning, and retrieval-based alignment—aren't new.
This contribution lies in combining these elements for cross-modal alignment,
which is more of an incremental step than a truly new idea.
Prior work such as OSCAR, VinVL, and LLaVA already explored fine-grained alignment using object tags or language models, and recent approaches like BLIP-2 and FLAVA address similar goals without slot-based decomposition.
This current paper lacks strong theoretical or empirical evidence showing that slot-level alignment is superior to token-level cross-attention or prototype-based clustering, making its originality less persuasive.

## Quality
This experimental design shows improvement over ViLT, OGM, and TSM-AV, but these models are now outdated compared to current open source SOTA like LLaVA, and Llama-3.
Without comparisons to these stronger baselines, it is unclear whether the proposed method is competitive in modern settings.
The current evaluation ignores more challenging benchmarks like instruction-following, which are key to recent multi-modal research.
This paper also lacks quantitative metrics for slot-level consistency beyond visualization, which makes the claims less rigorous, and limits the rigor of their claims about fine-grained alignment.

## Clarity
While this paper explains the pipeline and provides mathematical formulations, the discussion of limitations and alternative approaches is too brief.
While the role of adaptive slot selection and its impact on performance is described,
the reasons for design choices—like why slots beat token-level alignment—aren't deeply analyzed.
The absence of a clear comparison to alternative mechanisms like cross-attention or mixture-of-experts makes it harder to understand the unique benefits of the proposed approach.

## Significance
The idea of leveraging pre-trained uni-modal models to improve multi-modal learning is important,
but its significance is reduced by the lack of evidence that it scales to current architectures or tasks.
Improvements over older baselines are consistent but modest, and the paper does not show whether SGA can outperform or complement leading models that already achieve strong results without slot-based alignment.
To increase impact, authors are encourage to evaluate SGA on competitive benchmarks and provide a comparative analysis against alternative alignment strategies, demonstrating clear advantages in efficiency, interpretability, or data efficiency.

**Questions:**

Q1. What is the fundamental difference between SGA and existing approaches that use cross-attention or prototype-based clustering for fine-grained alignment?

Q2. Could you provide a theoretical or empirical justification that slot-level decomposition offers advantages beyond interpretability? For example, why should slots be better than token-level alignment in representation quality or efficiency?

Q3. How does the method perform under very low-resource conditions compared to strong models like CLIP or BLIP? Would running few-shot or zero-shot experiments on modern benchmarks help support this claim?

Q4. Why did you choose older baselines like ViLT, OGM, and TSM-AV instead of newer models such as LLaVA or Llama-3? Do you expect SGA to work easily with these new architectures? If so, can you show or explain why it should scale to current SOTA systems?

Q5. Can you provide data showing the impact of adaptive selection compared to just using a fixed number of slots? For instance, how much of the total performance improvement comes from adaptive slot selection?

Q6. Have you thought about including metrics like alignment accuracy against ground-truth region-to-phrase correspondences or IoU-based evaluations to support your claim of fine-grained alignment?

Q7. Why did you choose this over more common alternatives like InfoNCE or Sinkhorn-based approximations?

Q8. Have you compared its computational cost and stability to those alternatives?

Q9. How much does the proposed framework depend on the specific choice and number of pre-trained uni-modal models? The current ablation suggests shows stronger models help, but can you offer any guidance on diminishing returns or trade-offs if we keep adding more uni-modal sources?

---

### Official Review · Reviewer_yDKB · 2025-11-01

**Soundness:** 2
**Presentation:** 3
**Contribution:** 2
**Rating:** 2
**Confidence:** 4

**Summary:**

Many multimodal models only learn coarse links like image-caption because fine-grained supervision (region-word, frame-sound) is expensive. Meanwhile, strong unimodal models such as ViT or BERT already encode rich, disentangled concepts. The paper asks whether this existing unimodal structure can be transferred into a multimodal model trained on weakly aligned data.

It introduces Slot-Guided Fine-Grained Alignment (SGA), both the multimodal model and the frozen unimodal teachers run their features through adaptive slot attention to produce small sets of concept slots. The method then finds the best-matching slot pairs across models and uses a contrastive loss to pull those pairs together while pushing other slots apart; a standard retrieval loss keeps whole paired samples close. This yields concept-level alignment without detectors or manual region labels.

When added to existing models like ViLT, SGA improves image-text retrieval, VQA, and audio-visual tasks, with the biggest gains appearing when multimodal data is limited or when the unimodal teachers are stronger, showing that it is effectively distilling their knowledge.

**Strengths:**

- The idea of aligning concept-level slots from a multimodal model with slots from strong, frozen unimodal teachers is interesting
- Because the losses are add-on style, in principle you could add SGA onto a better backbone

**Weaknesses:**

- Preserving unimodal structure in the multi-modal representation space have previously been discussed by works such as Sirnam et. al. [1]. This related work is missing and not discussed. The paper would benefit from a clearer comparison and positioning of SGA relative to prior structure-preserving approaches.

- ViLT itself is from early 2021, with a very minimal visual backbone, recent progress in the field has gone much further. Many recent works like InternVL 2.x/2.5, LLaVA-NeXT/OneVision, Florence-2 variants, and PaliGemma 2 approach 60-70%+ on MMMU, DocVQA-style, chart/ocr benchmarks

- Applying the proposed method on a modern VLM would be much more interesting.

- SGA assumes you have good unimodal teachers. But in 2025, the best “teachers” are themselves multimodal


[1] Preserving Modality Structure Improves Multi-Modal Learning. Sirnam Swetha, Mamshad Nayeem Rizve, Nina Shvetsova, Hilde Kuehne, Mubarak Shah. ICCV 2023

**Questions:**

1. Authors argue SGA gives concept-level alignment, but evaluation is mostly classic VL tasks (COCO/Flickr30k retrieval, VQA, NLVR2) that can often be solved with coarse alignment. Can you report a task where coarse image-text contrastive training fails but SGA succeeds, to more directly validate the “fine-grained” claim ?

2. For slot selection, they use competitive top-K matching. How sensitive are results to K, to the slot-attention capacity, and to noise in the similarity matrix ? Did you try softer (Sinkhorn/Optimal Transport) matching, and if so, why was hard selection preferred?

3. Your core experiments are on top of ViLT, which is a 2021 early-fusion model and no longer representative of current VLM practice, do you expect the same slot-guided gains on stronger encoders/towers ? Can you provide some results to demonstrate this ?

---

### Official Review · Reviewer_7LK9 · 2025-11-02

**Soundness:** 3
**Presentation:** 2
**Contribution:** 2
**Rating:** 2
**Confidence:** 4

**Summary:**

The paper proposes Slot-Guided Alignment (SGA), a method for enhancing multi-modal learning by aligning it with knowledge from pre-trained uni-modal models. The core strength of the work is its novel and general framework that demonstrates strong performance and data efficiency. However, the review identifies significant concerns regarding clarity, methodological justification, and experimental completeness.

**Strengths:**

Interesting Framework: The proposed SGA method is a creative approach to a key problem (the modality gap) by leveraging the vast "ecosystem" of existing uni-modal models. Its ability to work across diverse modalities (image, text, audio, video) is a significant advantage.
Data Efficiency: The demonstrated ability to achieve high performance with fewer data samples is a highly valuable and practical contribution, making it relevant for data-scarce scenarios.
Strong Empirical Results: The paper shows convincing experimental results across multiple tasks and modalities, where SGA consistently improves baseline performance.

**Weaknesses:**

Unclear Terms: The term "concept" is used repeatedly but is never clearly defined. It is ambiguous whether it refers to latent model representations or human-interpretable semantic concepts, which undermines the clarity of the method's contribution.
Insufficient Methodological Justification: The rationale for choosing slot attention over other, potentially simpler decomposition techniques (e.g., additive models) is not provided. The authors proceed to decompose the latent space of joint models and uni models into slot spaces. Again the word of concept is introduced without sharing in any prior section which makes it unclear what does concept represent ? Latent concepts or human annotated concepts ?
While I like the idea in Sec 3.3 let me ask the authors why specific slot attention model ? Perhaps as a simple way I could decompose the latent space of the multimodal model (H) and unimodal models (h_m) into something like a additive model (H=H_u+H_n) and same for (h^m= h^m_u+h^m_n). My implicit assumption is that the latent space can be decomposed as an additive model and I can align the important parts say H_u and h^m_u and maximise the distance between H_n and h^m_n akin to contrastive learning. I dont understand the advantage of slot attention in this case especially since you are using aligned and unimodal data but with my assumption perhaps I can find the representations where actually representation might be important.

Incomplete Experimental Analysis: A critical piece of analysis is missing: a direct comparison of the performance of the powerful pre-trained uni-modal models against the final multi-modal model. This is needed to properly contextualize the "knowledge transfer" and demonstrate that the multi-modal model is indeed surpassing its teachers. To me the key piece of cross modal alignment or paired data is required contrary to what abstract states

Technical Inconsistencies: There are minor but noticeable inconsistencies in notation (e.g., d={x}_{I=1}^{M} vs. {X}_{I=1}^{M}) that should be corrected for professionalism and clarity.

**Questions:**

1)How do you precisely define a "concept" in the context of your slots? Is it a human-interpretable semantic unit or a purely latent, statistical feature within the model's representation?
2)Could the authors clarify why was slot attention chosen as the decomposition mechanism? Were other, simpler alignment techniques explored? What is the specific advantage of slot attention in this context that justifies its use? Perhaps also clarify Eq 4 where the authors mention optimal transport distance but the equation looks like cosine similarity. Am I missing information ? Elucidate this.
3) Can you provide a performance comparison between the pre-trained uni-modal models used and your final SGA-enhanced multi-modal model? This would help demonstrate the extent and value of the knowledge being distilled.
4)The paper highlights the cost of paired data, yet the method seems to require a paired multi-modal dataset d for alignment. Can you clarify the specific data requirements and how this reconciles with the stated motivation?

---

### Note · Authors · 2025-11-20

**Comment:**

Thanks for your all suggestions.

**Withdrawal Confirmation:**

I have read and agree with the venue's withdrawal policy on behalf of myself and my co-authors.